# Female vulnerability to the effects of smoking on health outcomes in older people

**Amin Haghani**[1]*, **Thalida Em Arpawong**[1], **Jung Ki Kim**[1], **Juan Pablo Lewinger**[2], **Caleb E. Finch**[1], **Eileen Crimmins**[1]

1 Leonard Davis School of Gerontology, University of Southern California, Los Angeles, California, United States of America, 2 Department of Preventive Medicine, University of Southern California, Los Angeles, California, United States of America

* ahaghani@usc.edu

**Data Availability Statement:** Data are available on re3data: https://www.re3data.org/repository/r3d100010862.

## Abstract

Cigarette smoking is among the leading risk factors for mortality and morbidity. While men have a higher smoking prevalence, mechanistic experiments suggest that women are at higher risk for health problems due to smoking. Moreover, the comparison of smoking effects on multiple conditions and mortality for men and women has not yet been done in a population-based group with race/ethnic diversity. We used proportional hazards models and restricted mean survival time to assess differences in smoking effects by sex for multiple health outcomes using data from the U.S. Health and Retirement Study (HRS), a population-representative cohort of individuals aged 50+ (n = 22,708, 1992–2014). Men had experienced more smoking pack-years than women (22.0 vs 15.6 average pack-years). Age of death, onset of lung disorders, heart disease, stroke, and cancer showed dose-dependent effects of smoking for both sexes. Among heavy smokers (>28 pack-years) women had higher risk of earlier age of death (HR = 1.3, 95%CI:1.03–1.65) and stroke (HR = 1.37, 95%CI:1.02–1.83). Risk of cancer and heart disease did not differ by sex for smokers. Women had earlier age of onset for lung disorders (HR = 2.83, 95%CI:1.74–4.6), but men risk due to smoking were higher (Smoking-Sex interaction P<0.02) than women. Passive smoke exposure increased risk of earlier heart disease (HR = 1.33, 95%CI:1.07–1.65) and stroke (HR:1.54, 95%CI:1.07–2.22) for non-smokers, mainly in men. Smoking cessation after 15 years partially attenuated the deleterious smoking effects for all health outcomes. In sum, our results suggest that women are more vulnerable to ever smoking for earlier death and risk of stroke, but less vulnerable for lung disorders. From an epidemiological perspective, sex differences in smoking effects are important considerations that could underlie sex differences in health outcomes. These findings also encourage future mechanistic experiments to resolve potential mechanisms of sex-specific cigarette smoke toxicity.

## Implications

This study identifies new sex differences in health outcomes due to smoking exposures. Sex differences in smoking hazards are often missed in epidemiological studies. Specifically,

**Funding:** This research is supported by findings from Cure Alzheimer's Fund (Caleb Finch) and National Institute on Aging of United States: Amin Haghani (T32AG052374, Kelvin Davis); Caleb Finch (R01AG051521, P01-AG055367, and P50AG05142-31).

**Competing interests:** The authors have no conflict of interest to declare.

interactions between sex and smoking dosage have not been fully examined with respect to differences in health outcomes in aging. Several rodent studies have evaluated sex differences in responses to cigarette smoke to support further examination of these interactions in humans. Critical gaps for future studies include sex differences in smoking responses at different life stages (development vs adulthood vs aged); and the molecular mechanisms of sex interaction with cigarette smoke.

## Introduction

Smoking is a leading cause of global mortality (6.5 million excess deaths) [1]. While men exhibit higher prevalence of smoking compared to women, studies have shown that women smokers have worse outcomes. For instance, women smokers with lung cancer show higher DNA adducts and mutation in P53 gene [2]. Sex-specific biological effects of cigarette smoke are supported by experimental studies in mice. Chronically exposed female mice showed significantly greater deficits than males in lung airway remodeling, increases in biomarkers for oxidative stress, inflammation, [3] and allergic reactivity [4]. In epidemiological studies, female smokers have shown higher risk for coronary heart disease [5], stroke [6], lung cancer [2, 7], bladder cancer [8], and chronic obstructive pulmonary disease [9]. However, epidemiological findings are mixed [10, 11]. Such discrepancies may be attributable to differences in the method of quantifying smoking exposure, use of cross-sectional data, small sample sizes, and use of younger cohorts when studying aging-related diseases.

Smoking hazards are dose-dependent [12], but we lack clear information on sex differences in the dose-dependence [8]. Some studies have found excess risk for men in all-cause mortality when cigarette dosage was quantified as the number of current cigarettes smoked per day [13]. In contrast, other reports on both sexes did not show a clear sex difference [14, 15].

Gender-specific health outcomes are even more obscure for individuals exposed to second-hand smoke, which is estimated to cause 650,000 deaths globally [12]. The few studies examining sex differences in passive smoking effects are inconsistent. For example, men were more vulnerable than women to passive smoking effects on risk of stroke in one study [16], whereas the opposite was reported in others [17, 18].

The current study examines the sex-specific smoking effects on aging-related health outcomes in the U.S. Health and Retirement Study (HRS), a large nationally representative U.S. aging study that has surveyed participants for more than 22 years. The comparison for smoking effects on males and females across multiple conditions and mortality has not yet been done in a population-based group with race/ethnic diversity. To more precisely estimate smoking dosage (vs. number of current cigarettes smoked per day), we calculated a pack-years index to represent the lifetime smoking exposure for each individual. We also examined passive smoking impacts on non-smokers with ever smoking spouses. We focus on how sex alters smoking hazards on the age of death and the onset of lung disorders, heart disease, stroke, and cancer. We further discussed the benefits of smoking cessation for different health outcomes. Evaluating these multiple outcomes allows us to examine the disease-specificity of smoking and the sex interaction. Potential biological mechanisms are discussed for interpretation of findings on sex and disease specificity.

## Methods

### Study population

Participants were a part of the 1992–2014 waves of the HRS, which is a nationally representative, longitudinal study of health and aging in the United States including adults (50+) and their spouses [19]. HRS is a publicly available data and no new data was collected for our

analysis. All HRS participants gave their consent to enter the study. The current analysis used 12 waves of data, collected every two years from 1992 through 2014. Respondent information was obtained from the RAND 2014 HRS datafile, in addition to HRS Core data files for each wave. The data were comprised of five HRS cohorts (the original HRS cohort born 1931–41, Children Of the Depression born 1924–30, War Babies born 1942–47, Early Baby Boomers born 1948–53, and Mid Baby Boomers born 1954–59), which are recruited in the study in years 1992, 1998, 2004, or 2010 at ages 51–61 (S1–S3 Figs). The older AHEAD cohort was excluded from the analysis due to its entry into HRS after age 70. Only 122 individuals from AHEAD cohort who entered between ages 51–61 were included in the study. The original HRS cohort has up to 22 years of follow-up and the Mid Baby Boomers had up to 4 years of follow-up, which are the longest and shortest average years of follow-up in the cohorts (S1 Fig).

## Outcome variables

Age of death was computed from the year of death variable in the RAND file (radyear), which is based on the National Death Index and exit interviews with proxy respondents. Two variables were constructed for each of the health conditions: prevalence, which is a binary variable for having been diagnosed with the condition; and age of onset, which is the earliest reported age of the health problem. The incidence of specific diseases was based on the question "whether or not a doctor has told the respondent that s/he had these conditions". The age of onset was extracted from the responses to "In what year (when) did you have or were diagnosed with the condition". For individuals with no prior history of the condition, the age at the wave of incidence was considered as the first age of onset for the condition. The health conditions include: 1) Lung disorders including chronic bronchitis and emphysema but not asthma; 2) Heart disease including heart attack, coronary heart disease, angina, congestive heart failure or other heart problems; 3) Stroke or transient ischemic attacks (TIA); and 4) Cancer which included any kind of cancer or malignant tumor, except for skin cancer.

## Predictor variables

Lifetime exposure to smoking is indexed as lifetime pack-years smoked. The pack-year variable is calculated as the multiplicand of reported average number of cigarette packs smoked daily by lifetime years of smoking. Briefly, the earliest age of smoking was extracted from responses to "How many years ago", "what year", or "what age did you start smoking?" The age of smoking cessation was extracted from questions on "How many years ago", "what year", or "what age did you stop smoking?" The earliest age reported for starting and the latest age for cessation were used for each individual to calculate total years of smoking (S4 Fig). Cigarettes smoked per day were calculated from both the average of the reported number of cigarettes per day at each wave for each individual, and the maximum number of cigarettes smoked during the time in which the individual reported smoking the most (S4 Fig). Around 10,123 ever smokers (44%) had at least one missing value for calculating pack-years. These individuals included ever smokers with no reported age of start, former smokers with unknown age of cessation, and ever smokers who did not report the number of cigarettes per day. Since this was a large portion of the population, missing values were imputed using the average age of starting, age of cessation, and number of cigarettes per day calculated from the 12,585 ever smokers who had complete data. S5 Fig shows the number of individuals with or without data imputation in the analysis. The continuous pack-year variable (multiplicand of average daily packs and years of smoking) was then classified into dosage quartiles for analysis. The reported results are based on the whole data (imputed and non-imputed), however, sensitivity analysis confirmed the same pattern of findings in the sub-population with no imputation (S2 Table).

Passive smokers are defined as never smokers who lived with at least one smoker spouse. The difference between the age of smoking cessation and the latest age in the study was defined as years since smoking cessation in former smokers. This variable was converted to a categorical variable as follows: <5, 5–15, >15 years of cessation.

The demographic characteristics for sex, race (White/Caucasian, Black/African American, Other), and ethnicity (Hispanic/non-Hispanic) were extracted from the HRS RAND file. The ethnicity variable was constructed from the self-reported race and ethnicity as follows: White (non-Hispanic White), African American (non-Hispanic African American), Hispanic, and Other (non-Hispanic others).

### Statistical analysis

Hazard ratios (HRs) for sex, smoking pack-years, passive smoking were calculated using Cox proportional hazard modeling for age of death, and onset of health conditions. The models estimated time after age 50 to event, and included an interaction term for sex and smoking to evaluate differences between men and women. All models were adjusted for ethnicity. The HRs were also calculated in sex-stratified models for ever and passive smoking effects by sex. The restricted mean survival time (RMST) of each group was calculated from the Cox proportional hazard model. Survival curves were fitted for the sex-stratified data on the Cox models that included a sex-smoking interaction term to estimate RMST of each group. The RMST can be interpreted as the average of event-free survival time from 50 to 85 years old age that is adjusted for ethnicity [20–22]. The analysis was done in R (version 3.5.3), using the survival package. The Cox-proportional hazard formula is:

$$h(t) = h_0(t)\exp(\beta 1 X 1 \ldots + \beta p X p)$$

where $h(t)$ represents expected hazard at age $t$; the $h0(t)$ is the baseline hazard when all of the predictors are 0; β, coefficients; X, the predictors which included sex, ethnicity, different categories of pack-years (or passive smoke), and sex interaction with each pack-year categories (or passive smoke).

S3 and S4 Tables summarize the results of cox proportional hazard models with additional controls for other confounders including years of education, and cohort. Adjusting for these confounders did not affect the results on the smoking hazards.

## Results

Demographics of the HRS sample with pack-year categories are in S1 Table. The 22 years of the study included 22,708 age-eligible individuals, ages 50–85 years. Men and women had similar age (mean baseline age, 66) and were balanced for most variables, with exceptions of a female excess for passive smokers (10.7% of women vs. 6.3% of men), non-smokers (26.8% of women vs. 19.7% of men), and medium smokers with 15–20 pack-years history (20.0% of women vs. 15.0% of men). Men had a greater percentage of very high smokers with >28 pack-years (25.7%) than women (13.3%). The health conditions with the highest and lowest prevalence were heart disease (26.0% in men, 22.0% in women) and lung disorders (5.9% in men, 7.0% in women).

### Dose-dependent smoking hazard ratios (HR)

Ever smokers had consistent dose-dependent HR for earlier death, and earlier onset of lung disorders, heart disease, and stroke for both men and women. Smoking-related HR for risk of death and lung disorders were elevated even at the lowest smoking levels (0.03–15 pack-years) (Table 1). The highest HR from ever smoking was observed for lung disorders, which ranged

**Table 1. Hazard ratios for age of death, and age of onset of lung disorders, heart disease, stroke and cancer according to lifetime smoking level and the interaction with sex.**

| variable | level HR (95%CI) | Age of death | Lung disorders | Heart disease | Stroke | Cancer |
|---|---|---|---|---|---|---|
| Sex | Men (ref) | | | | | |
| | Women | **0.65 (0.52,0.8)**\*\*\* | **2.83 (1.74,4.6)**\*\*\* | **0.82 (0.71,0.95)**\*\* | 0.81 (0.64,1.02) | 1.14 (0.97,1.34) |
| ethnicity | White/Caucasian (ref) | | | | | |
| | African American | **1.64 (1.53,1.77)**\*\*\* | 0.93 (0.8,1.07) | 1.04 (0.97,1.12) | **2.06 (1.87,2.27)**\*\*\* | **0.86 (0.79,0.94)**\*\* |
| | Hispanic | 1.04 (0.93,1.15) | **0.79 (0.65,0.97)**\* | **0.8 (0.73,0.88)**\*\*\* | **1.34 (1.16,1.54)**\*\*\* | **0.73 (0.66,0.82)**\*\*\* |
| | Other | 1.05 (0.86,1.28) | 1.34 (0.99,1.81) | 1.03 (0.87,1.21) | 1.17 (0.89,1.54) | **0.73 (0.59,0.91)**\*\* |
| Pack years | Non-smokers (ref) | | | | | |
| | Low | **1.38 (1.14,1.67)**\*\* | **3.03 (1.85,4.96)**\*\*\* | 1.1 (0.95,1.27) | 1.21 (0.96,1.53) | 1.17 (0.99,1.39) |
| | Medium | **1.45 (1.2,1.75)**\*\*\* | **3.07 (1.88,5)**\*\*\* | **1.22 (1.06,1.41)**\*\* | 1.24 (0.98,1.56) | **1.22 (1.03,1.44)**\* |
| | High | **1.43 (1.2,1.71)**\*\*\* | **3.34 (2.09,5.35)**\*\*\* | **1.33 (1.16,1.51)**\*\*\* | **1.24 (1,1.54)**\* | 1.12 (0.96,1.31) |
| | Very high | **2.24 (1.89,2.66)**\*\*\* | **6.95 (4.42,10.92)**\*\*\* | **1.52 (1.34,1.73)**\*\*\* | **1.52 (1.23,1.87)**\*\*\* | **1.25 (1.07,1.45)**\*\* |
| Women x Smoking interaction | Low | 1.1 (0.85,1.43) | **0.44 (0.25,0.78)**\*\* | 1.19 (0.98,1.45) | 0.96 (0.7,1.31) | 0.87 (0.69,1.08) |
| | Medium | 1.02 (0.79,1.31) | **0.42 (0.24,0.73)**\*\* | 0.95 (0.79,1.15) | 1.01 (0.75,1.38) | **0.79 (0.64,0.98)**\* |
| | High | 1.17 (0.92,1.49) | **0.45 (0.26,0.76)**\*\* | 0.9 (0.75,1.08) | 1.1 (0.83,1.47) | 0.86 (0.7,1.06) |
| | Very high | **1.3 (1.03,1.65)**\* | **0.56 (0.34,0.94)**\* | 1.07 (0.89,1.28) | **1.37 (1.02,1.83)**\* | 0.98 (0.79,1.2) |
| Total N | | 22708 | 21486 | 22708 | 22695 | 22689 |

\* p < 0.05,

\*\* p < 0.01,

\*\*\* p < 0.001

from 3.0 (95%CI 1.85–4.96) to 7.0 (95%CI 4.42–10.92), with HRs progressively increasing with higher smoking dosage. Increase in smoking dosage from 15 to >28 pack-years caused a 20% increase in the HR of earlier onset for heart disease and stroke: an HR = 1.2 (95%CI 1.06–1.41) to 1.5 (95%CI 1.34–1.73). While smoking increased the risk of earlier onset of cancer (1.2, 95% CI 1.07–1.45), there was no clear pattern of dose-dependence.

## Sex-specific ever smoking hazards

Overall, women died at older ages than men and had later onset of specific health conditions (Table 1). Women had a lower HR for earlier death (HR = 0.65, 95%CI 0.52–0.8) and earlier diagnosis with heart disease (HR = 0.82, 95%CI 0.71–0.95). In contrast, women had a higher HR for earlier onset of lung disorders than men (HR = 2.83, 95%CI 1.74–4.6).

Gender interactions with ever smoking varied by outcome and smoking dosage. For very heavy smoking (> 28 pack-years) women had a higher HR for earlier death (HR = 1.3, 95%CI 1.03–1.65) and earlier stroke (HR = 1.37, 95%CI 1.02–1.83) than men (Fig 1A). In contrast, women smokers showed a lower risk of earlier lung disorders diagnosed than men smokers, particularly in low, medium and high smokers (Fig 1B). As noted above, women had a higher main effect of lung disorders (HR = 2.8, 95%CI 1.74–4.6) than men. Level of smoking and sex did not show a strong interaction on the outcomes of earlier onset for heart disease or cancer. The sensitivity analysis in non-imputed smoking data showed similar but stronger sex interactions with smoking hazards (S2 Table). Thus, in the main text, we describe the observed hazards in the full sample.

The order of conditions based on earlier RMST age (an indicator of average event-free age), is as follows: heart disease (men, RMST age 74.2; women, 75.3), cancer (men, 77.8; women, 77.1), death (men, 77.7; women, 79.3), stroke (men, 80.5; women, 82), and lung disorders

### A. Hazard ratio of ever smoking on age of death

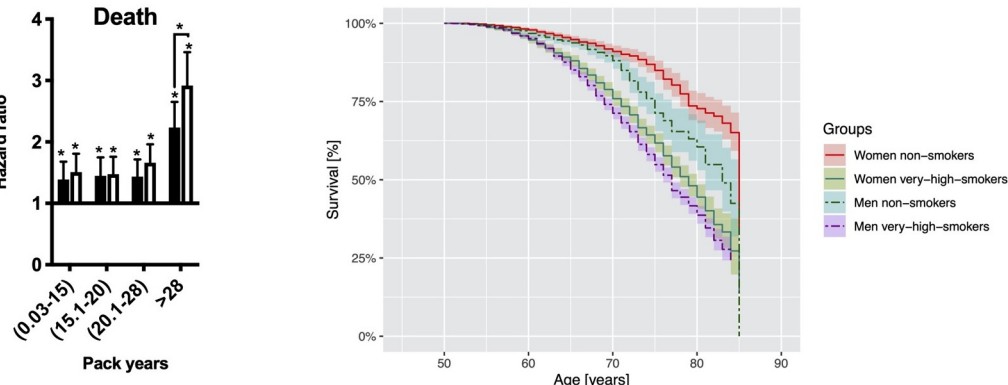

### B. Hazard ratio of ever smoking on age of onset of health outcomes

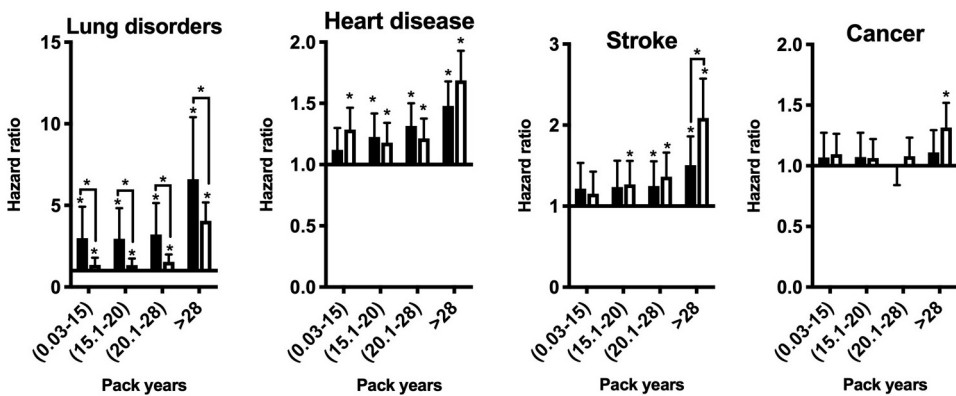

### C. Hazard ratio of passive smokingon age of onset of health outcomes

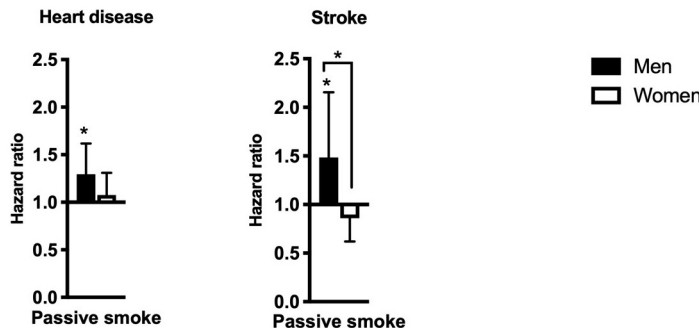

**Fig 1. Smoking hazards are modified by sex for earlier age of death, and earlier onset of lung disorders, and stroke.**
A) The smoking pack-years hazard and Kaplan-Meier survival curves for all-cause mortality in men and women between 50–85 years old age in HRS. The hazard ratio (HR) of smoking pack-years (B) and passive smoke (C) on health outcomes. HRs are calculated using Cox proportional hazard models separately by sex. *p-value <0.05 based on Wald test in the models. The significant sex differences are based on the interaction terms in the full model and are indicated in each figure. The stratified models were adjusted for ethnicity. The baseline effects of sex are reported in Tables 1 and 2.

(men, 81.1; women, 79.2) (Fig 2A). RMST analysis showed that very high smoking (>28 pack-years) has greater impact on women than men for earlier age of death (4.7 years earlier for women, 4.1 years earlier for men), and stroke (2 years earlier for women, 1.1 years earlier for men). Men showed greater vulnerability to smoking for the earlier age of onset for lung

**Fig 2. Restricted mean survival time (RMST) of death and disease onset in men and women smokers.** The calculated RMST is based on Cox proportional hazard model of (A) smoking pack-years, and (B) passive smoke. The results are reported as RMST±SEM. The lines are showing the number of years that was reduced by smoking from the baseline RMST age in each sex.

disorders, particularly in smokers with less than 28 pack-years (women, -0.8 years; men, -1.3 years). Very heavy smoking had similar effects on the earlier onset of heart diseases and cancer in both sexes (Heart diseases: women, -3.3 years, men, -3.1 years; Cancer: women -1.6 years, men -1.5 years).

## Passive smoking hazards

Passive smoking increased the risk of earlier onset of heart disease (HR = 1.3, 95%CI 1.07–1.65) and stroke (HR = 1.5, 95%CI 1.07–2.22) (Table 2). Age of death and age of onset of lung disorders, and cancer were not affected by passive smoking. In HRS, hazards of passive smoke were close to those associated with very high smoking exposure for earlier onset for heart disease (HR Passive/Ever, 1.33/1.52; RMST Passive/Ever 77/75) and stroke (HR Passive/Ever, 1.54/1.52; RMST Passive/Ever, 82/80).

Based on Cox proportional hazard models, passive smoking effects were modified by sex only for earlier age of stroke. Only passive smoking males showed an increase in the risk of earlier age of stroke compared to females (Fig 1C). The RMST analysis showed 2 years earlier onset for heart diseases (at around age 75), and 1.1 years earlier onset for stroke (at age 81) only in male passive smokers (Fig 2B). Heart disease and stroke onset did not show any baseline differences between male and female never smokers with non-smoker spouses.

**Table 2. Hazard ratios of age of death, and age of onset of lung disorders, heart disease, stroke and cancer according to passive smoke and the interaction with sex.**

|  | HR (95%CI) | Age of death | Lung disorders | Heart disease | Stroke | Cancer |
|---|---|---|---|---|---|---|
| Sex | Men (ref) |  |  |  |  |  |
|  | Women | 0.73 (0.54,0.98)* | **2.92 (1.51,5.65)**\*\* | 0.89 (0.73,1.08) | 1.02 (0.74,1.41) | **1.33 (1.07,1.65)**\* |
| Ethnicity | White/Caucasian (ref) |  |  |  |  |  |
|  | African American | **1.37 (1.05,1.8)**\* | 1.5 (0.95,2.39) | **1.2 (1,1.43)**\* | **2.93 (2.24,3.82)**\*\*\* | **0.8 (0.64,0.99)**\* |
|  | Hispanic | 1.03 (0.75,1.42) | 1.22 (0.72,2.05) | **0.72 (0.57,0.9)**\*\* | **1.48 (1.05,2.09)**\* | **0.76 (0.6,0.96)**\* |
|  | other | 0.73 (0.37,1.42) | 1.03 (0.37,2.83) | **0.61 (0.39,0.96)**\* | 1.16 (0.59,2.28) | **0.5 (0.3,0.84)**\*\* |
| Smoking | Never smokers (ref) |  |  |  |  |  |
|  | Passive Smokers | 1 (0.72,1.38) | 1.22 (0.5,2.95) | **1.33 (1.07,1.65)**\* | **1.54 (1.07,2.22)**\* | 1.26 (0.97,1.64) |
| Women x passive smoking interaction |  | 0.71 (0.46,1.1) | 0.86 (0.32,2.29) | 0.78 (0.58,1.05) | **0.52 (0.32,0.84)**\*\* | **0.69 (0.5,0.96)**\* |
| Total N |  | 5309 | 5201 | 5309 | 5305 | 5298 |

\* p < 0.05,

\*\* p < 0.01,

\*\*\* p < 0.001

## Smoking cessation and health outcomes

Because ever smokers include both active and former smokers, we conducted sensitivity analysis based on smoking status to examine the hazards of smoking and potential sex differences. Former smokers were categorized by the number of years since cessation. Former smokers with >15 years of cessation recovered from ever smoking hazards for earlier onset of stroke (HR = 0.89, 95%CI 0.79–1.10) and cancer (HR = 1.08, 95%CI 0.93–1.25) (S5 Table). This group of former smokers (>15 years cessation) were still at higher risk for lung disorders (HR = 2.67, 95%CI 1.68–4.23) and heart disease (HR = 1.22, 95%CI 1.08–1.39) than the non-smokers. Nonetheless, they were at lower risk compared to current smokers. Smoking cessation of >15 years also showed protective effects against mortality compared to non-smokers (HR = 0.48, 95%CI 0.4–0.57) (S5 Table, Fig 3A).

Former smokers with fewer than 15 years of cessation showed increased risks for all health outcomes compared to current smokers: earlier death (HR for current smokers vs. cessation <5 years: 1.29 vs 6.26); lung disorders (4.3 vs 8.24); heart disease (1.29 vs 1.5); stroke (1.32 vs 2); cancer (1.14 vs 1.55). Former smokers with more than 15 years since cessation, on average stopped smoking around age 55 (Fig 3B), which was at least 5 years earlier than the former smokers who reported 10–15 years of cessation.

This sensitivity analysis did not enable us to detect sex differences in the health hazards of current smokers. The effects of smoking cessation itself were modest in that they differed by sex for only some health outcomes: earlier death and risk of lung disorders (S5 Table).

## Discussion

Our findings show dose-dependent sex differences in smoking for premature mortality and morbidity among a nationally-representative sample of U.S. adults. Women were more vulnerable to ever smoking for premature death and stroke incidence. In contrast, for <28 pack-years, men smokers had earlier lung disorders than women. The onset age of cancer and heart disease in smokers did not differ by sex, even for very heavy smokers. Passive smoke exposure indicated excess risk for men for earlier heart disease and stroke.

These findings for women are consistent with experimental studies. Mice chronically exposed to cigarette smoke for 6 months had greater responses among females than males for lung small airway remodeling and increased distal airway resistance. Corresponding

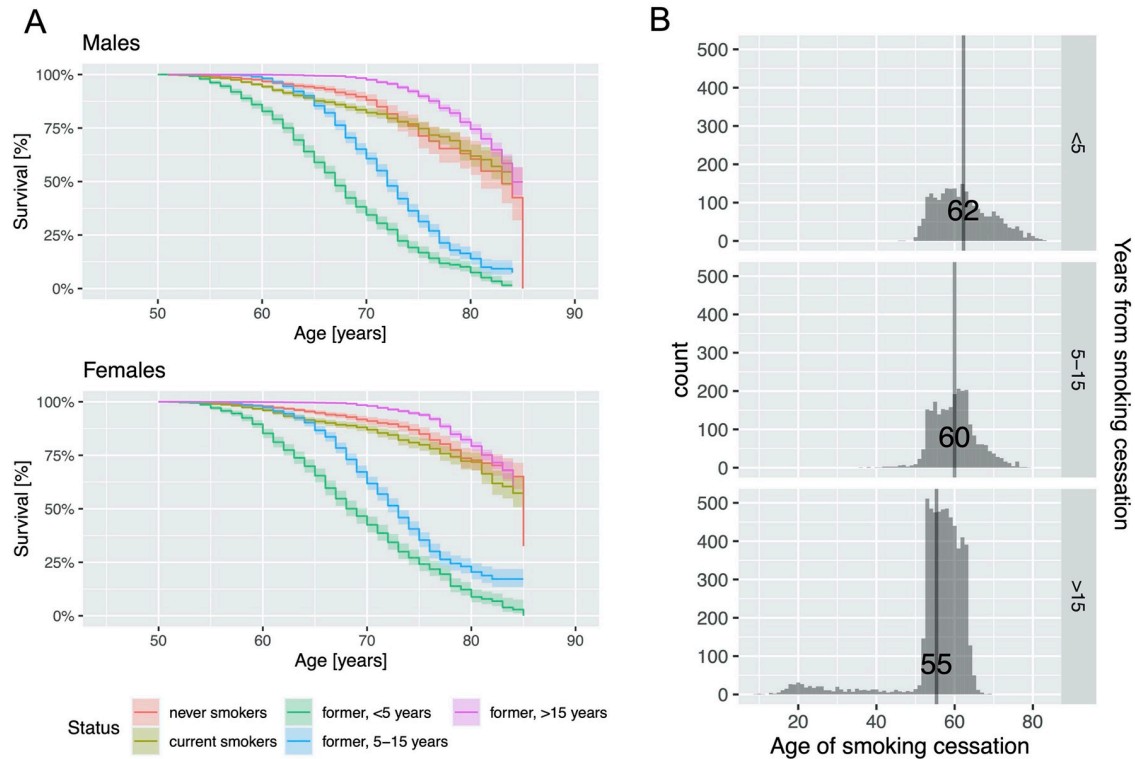

**Fig 3. Smoking cessation >15 years reduced the hazards of smoking on age of death in both men and women.** A) Kaplan-Meier survival curves for all-cause mortality in men and women between 50–85 years for smoking cessation status. B) Histogram and average of age of smoking cessation in different groups of former smokers in HRS.

biomarkers included 2-fold increase in 3-nitrotyrosine (oxidative stress) and 1.5-fold increase in transforming growth factor β (inflammation) [3]. In contrast, male mice of this study had higher induction of genes mediating oxidative stress responses (Nrf2, Nqo1), and detoxification (Cyp1a1, Cyp1b1). The role of sex steroids is shown by ovariectomy, which ameliorated lung airway remodeling [4]. In other studies replacement of 17βc-estradiol (E2) decreased autophagy and increased Nrf2 responses of hippocampus to cerebral ischemia [23]. Thus, steroid hormones could underlie some of these sex differences to cigarette smoke.

Analysis of older birth cohorts from 1800–1935 in US and European counties showed increasing excess adult mortality for men born after 1900 [24]. Around 30% of this excess in men's mortality was attributed to smoking; however, this analysis was restricted to using smoking status (smokers vs non-smokers), as pack-years were not known. From this analysis of HRS, we can infer that more of the excess male mortality observed in earlier historical cohorts reflects the smoking behaviors of men. As we show that men smoked 30% more pack-years than women (S1 Table, an average of 22.0 vs 15.6). Of key relevance, the current findings now provide a characterization of individual smoking exposure and mortality for those born after 1934.

In other studies, women smokers had 2-fold more DNA adducts or frameshift mutations in the P53 gene than men smokers [2]. This higher DNA damage among women smokers was also associated with a more accelerated risk of cancer and other age-related chronic diseases [25, 26]. While the current study showed that smoking increased risk for earlier age of cancer, it did not show a clear sex interaction. In sex stratified analysis, the risk of an earlier age of cancer onset was significant only for women smokers with very high pack-years, which is

consistent with the women's excess of mutations in lung cancer cited above [2, 7]. Future studies should examine sex-smoking interactions for specific cancer types.

Lung disorders in the current study included all chronic lung conditions other than asthma, such as chronic bronchitis and emphysema. Thus, the assessment of the outcome includes chronic obstructive pulmonary disease (COPD). North American regional studies have shown men excess of COPD from cigarette smoking [27–29], which parallels our finding of greater risk for earlier lung disorders onset in men smokers. These findings contradict those in mouse models, that found only female responses in airway remodeling due to chronic exposure [3]. However, these mouse models may not represent older human ages. One concern in interpreting these findings on lung conditions is the potential under-diagnosis of COPD in North America, particularly for women [30]. Further experimental studies may resolve the sex-specific cigarette smoke effects for the onset of lung disorders at later ages.

Hazards of passive smoke were comparable to very high pack-years for earlier onset for heart disease and stroke. Importantly, the chemical composition of side-stream (passive) smoke differs from main-stream (inhaled) smoke, with much lower density of particles and gases, by 10 to-100 fold [31]. However, per cigarette, side-stream smoke has 2- to 30-fold higher concentrations of nicotine and organic toxins (benz[a]pyrene, and other polycyclic aromatic hydrocarbons); volatile hydrocarbons (ethene, propene); and N-nitrosamines; and gases (carbon monoxide, nitric oxide) [31, 32]. Our recent studies on ambient air pollution particulate matter showed that chemical and physical characteristics of the particles can largely affect the toxicity of air pollution samples with the same mass concentration [33, 34]. Thus, it is not surprising that main-stream and second-hand smoke would diverge in toxicity and have different gender-specific effects. In HRS, men exposed to passive smoke had a higher risk of heart disease and stroke than women. In a mouse model of prenatal exposure to passive smoke, males had greater alteration in adult lung tidal volume [35].

Despite the well-documented magnitude of cigarette smoking hazards and decades of research on potential carcinogens and other toxins, we have still a surprisingly limited understanding of sex interactions. This analysis revealed that sex-specific smoking effects depend on the aging condition. The parallel experimental findings for sex differences in mice highlight the possibility of broadly shared biological mechanisms. Further population-level analyses are needed on sex differences in cigarette toxicity that may be shared with air pollution, including diseases of arteries, lungs, and brain [36]. For example, lung cancer risk scales with pack-years and air pollution levels of PM2.5 independently, while the combination has multiplicative synergies [37]. Further experimental studies of developmental and adult exposure to cigarette smoke could include mice with transgenes for detoxification gene variants associated with vulnerability to air pollution, e.g. alleles of the glutathione S-transferase gene GSTP1 [38] and MET receptor of tyrosine kinase [39].

In the last part of our analysis, we examined smoking hazards in former smokers by duration since cessation. These results suggested that smoking cessation before age 60 could attenuate the smoking hazards for all our target health outcomes. However, quitting smoking after age 60 was associated with additional stress and increased the risk of earlier death and other outcomes compared to current smokers. This aging effect could represent the declining regenerative capacity of many tissues after middle-age [40, 41]. Little is known of how smoking interacts with basic aging processes of cell senescence and systemic inflammation.

We must also consider whether spontaneous smoking cessation by older heavy smokers was due to their recognition of underlying diseases. Former smokers with less than 20 years of cessation had higher incidence of lung or prostate cancer than current smokers did [42–44]. In the Canadian National Breast Screening Study, which includes 49165 women aged 40–59,

those who quit smoking at older ages had a higher relative risk for lung cancer mortality than current smokers with high number of cigarettes per day or years of smoking [44].

Chronic damage from heavy smoking cannot be recovered from simply by cessation. An experimental study of male hypertensive rats modeled the long-term sequela of smoking (up to at age 10 months) and identified persistent damage effects that did not reverse 10 months after cessation [45]. Changes included higher mortality at age 21 months, and persistent inflammation in several lung regions. Examining the smoking cessation in HRS enabled us to characterize the complexity of smoking effects on health in the older population. However, we are not able to draw strong conclusions about sex differences in the benefits of smoking cessation. Understanding this relationship necessitates a larger cohort that allows the inclusion of both smoking status and smoking pack-years in one model. The benefits of smoking cessation could depend on several factors such as age, sex, smoking dosage, underlying disease, and other environmental risk factors.

Further studies of sex differences in response to cigarette smoke should consider physiologically distinct stages of the lifespan: development (0–18), young adulthood (18–35), middle-age including post-menopause (36–60), and older ages when chronic diseases increase exponentially (60+). For example, a mouse model of air pollution toxicity had attenuated responses of lung and brain by middle-age (18 months) [46, 47]. We should also consider potential middle-aged survivor bias in older age cohorts such as HRS, which combines individuals in late-middle age and older ages.

The current study examined the association of smoking and several aging-related conditions in a nationally representative dataset with follow-up over two decades and a robust measure for exposure to different smoking doses over a lifetime. Cigarette smoke effects may interact with diverse socioeconomic and environmental factors including birth cohort, diet and lifestyle, and exposure to air and noise pollution. In HRS, we showed that additional controls for years of education and birth cohort do not alter the smoking hazards or sex-specificity of the findings (S3 and S4 Tables). However, this conclusion is obscured by the complexity of the interaction between socio-economic status, cultural habits, genetic diversity, and cigarette smoke toxicity. Moreover, our results can be affected by the quality of self-reported smoking data. For example, former smokers reported a slightly higher history for the number of cigarettes per day than current smokers (0.5 higher average cigarettes per day, S4 Fig). It is unclear if such reporting captures true differences, or is due to reporting bias from former or current smokers. A series of controlled experiments can disentangle the contribution of these individual confounders on the smoking hazards and also contribute to the development of a biomarker that can estimate true smoking dosage, such as DNA methylation levels [48]. Future analyses of smoking effects on aging-related conditions should also consider interactions with other biological (e.g. age) and environmental (e.g. outdoor and indoor ambient air pollution) factors.

Despite the partially successful decrease of cigarette smoking in most developed countries, Asian and African markets for tobacco are still growing, which anticipates the need for new therapeutic and preventive measures with sex-specificity by age and life stage.

## Supporting information

**S1 Fig. Histogram of number of years of in different cohorts enrolled in HRS.**
(DOCX)

**S2 Fig. Histogram of year of birth of different cohorts enrolled in HRS.**
(DOCX)

**S3 Fig. Histogram of year of recruitment for each cohort in HRS.**
(DOCX)

**S4 Fig. Box plot of average number of cigarettes per day, smoking years, and pack years of 17,399 men and women current or former smokers in HRS.** The significance of difference between current and former smokers was assessed by t-test. * p<0.05.
(DOCX)

**S5 Fig. Histogram of the number of individuals with imputations used in creating the pack year variable.** In total, 12585 respondents had complete data, and 10123 respondents had at least one missing variable that had to be imputed (e.g. age of start smoking, age of quitting smoking, average daily cigarettes smoked).
(DOCX)

**S1 Table. Demographic characteristics of the HRS sample, 1992–2014.**
(DOCX)

**S2 Table. Hazard ratios of age of death, and age of onset of lung disorders, heart disease, and stroke according to ever smoking and the interaction with gender in data with no imputation.** This subpopulation showed a similar pattern but stronger sex-smoking interaction for the age of death, heart disease, and cancer. Thus, the main text reported the results of the whole population.
(DOCX)

**S3 Table. Hazard ratios of age of death, and age of onset of lung disorders, heart disease, and stroke according to ever smoking and the interaction with gender in data with no imputation.** Confounders: Years of education, Ethnicity.
(DOCX)

**S4 Table. Hazard ratios of age of death, and age of onset of lung disorders, heart disease, and stroke according to ever smoking and the interaction with gender in data with no imputation.** Confounders: Years of education, Cohort, Ethnicity.
(DOCX)

**S5 Table. Hazard ratios of age of death, and age of onset of lung disorders, heart disease, stroke and cancer according to years since quitting smoking and the interaction with sex.**
(DOCX)

## Author Contributions

**Conceptualization:** Amin Haghani, Thalida Em Arpawong, Caleb E. Finch, Eileen Crimmins.

**Data curation:** Amin Haghani.

**Formal analysis:** Amin Haghani.

**Funding acquisition:** Caleb E. Finch.

**Investigation:** Amin Haghani.

**Methodology:** Amin Haghani, Thalida Em Arpawong, Jung Ki Kim, Juan Pablo Lewinger, Eileen Crimmins.

**Project administration:** Thalida Em Arpawong.

**Resources:** Eileen Crimmins.

**Supervision:** Eileen Crimmins.

**Writing – original draft:** Amin Haghani.

**Writing – review & editing:** Amin Haghani, Thalida Em Arpawong, Caleb E. Finch, Eileen Crimmins.

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
