## [Decision Letter · Decision Letter 0]

18 Mar 2020

PONE-D-20-03538

Female vulnerability to the effects of smoking on health outcomes in older people

PLOS ONE

Dear Dr. Haghani,

Thank you for submitting your manuscript to PLOS ONE. After careful consideration, we feel that it has merit but does not fully meet PLOS ONE’s publication criteria as it currently stands. Therefore, we invite you to submit a revised version of the manuscript that addresses the points raised during the review process.

I agree with the reviewers that there are a number of strengths to the manuscript, but also some concerns that would need to be satisfactorily addressed before it could be accepted for publication. In particular, both reviewers' comments about providing more detail about the data and the analyses, and the recommendations to account for potential sources of bias, are important to address.

We would appreciate receiving your revised manuscript by May 02 2020 11:59PM. To enhance the reproducibility of your results, we recommend that if applicable you deposit your laboratory protocols in protocols.io, where a protocol can be assigned its own identifier (DOI) such that it can be cited independently in the future. For instructions see: http://journals.plos.org/plosone/s/submission-guidelines#loc-laboratory-protocols

We look forward to receiving your revised manuscript.

Kind regards,

Neal Doran

Academic Editor

PLOS ONE

Journal Requirements:

2. Please ensure you have thoroughly discussed any potential limitations of this study within the Discussion section, for example the impact of confounding factors.

Reviewers' comments:

Reviewer's Responses to Questions

**Comments to the Author**

1. Is the manuscript technically sound, and do the data support the conclusions?

Reviewer #1: Yes

Reviewer #2: No

2. Has the statistical analysis been performed appropriately and rigorously? 

Reviewer #1: Yes

Reviewer #2: No

3. Have the authors made all data underlying the findings in their manuscript fully available?

Reviewer #1: Yes

Reviewer #2: Yes

4. Is the manuscript presented in an intelligible fashion and written in standard English?

Reviewer #1: Yes

Reviewer #2: Yes

5. Review Comments to the Author

Reviewer #1: Thank you for giving me the chance to review this manuscript.

The study investigated the association between smoking and different health outcomes in a large population-based sample. Furthermore, sex-specific and dose-dependent smoking effects were evaluated. The paper is well written and contributes with interesting results to the previous literature. However, some points need to be clarified or could be discussed in more detail. Please see below for my specific comments.

Specific comments

• It could be mentioned in the abstract which statistical methods have been applied.

• In the abstract, HR 0.44-0.56 for the risk of lung disorders in women smokers is stated. This needs to be checked.

• What was the reason to exclude asthma? Only lung disorders such as chronic bronchitis and emphysema were considered.

• It is mentioned in the methods section that 12 waves of the Health and Retirement Study (1992-2014) were used for the current analyses. Does this imply that 12 follow-ups were available? Maybe some more information on the different waves and times of follow-up could be given.

• Would it be possible to specify the cause of death? Besides smoking there might be other risk factors which could lead to an early death. Have further potential confounding factors such as socio-economic status, comorbidities and other lifestyle-related factors been tested in the regression models?

• It is stated in the methods section that missing smoking values were imputed using average values. What was the proportion of imputed values? If the proportion was quite high it would be good to perform sensitivity analyses only including those with self-reported smoking information (i.e. exclusion of subjects with imputed values).

• The exact definition of the smoking categories is not entirely clear (presented in Table S1). The percentages for never, passive and active smokers sum up to 100% for men and women, respectively, meaning that these three groups are disjunctive. However, passive smokers are per definition also never smokers. Does this imply that the never smoker group comprises only never smokers without passive smoke exposure? It seems that former smokers are included in the active smoking category. Have it been investigated whether there are different health effects when differentiating former (e.g. quit smoking a long time ago) and current smokers?

• The resolution of the Kaplan-Meier curve is not sufficient as the legend is not readable.

• Potential biological mechanisms behind the sex-specific smoking effects could be discussed in more detail.

Reviewer #2: Comments to the Author:

This study examines sex-specific vulnerability to health outcomes from smoking at older ages in the HRS. They find that older women are more susceptible to earlier death and risk of stroke but are less likely to report a doctor-diagnosed lung disorder. Sex-specific incidence and comorbidity of smoking and other substance use disorders is critically under-examined in population studies, and the paper makes a novel and important contribution to this literature. However, I have some major and minor concerns that need to be addressed before I can recommend it for publication. These concerns are highlighted below.

Major Concerns:

My main concern is whether the study is well powered, particularly for the sex-specific findings for females in the “very high pack years” group who comprise only 13.3% of female smokers. These findings form the basis of the authors’ claim that older female smokers are more susceptible to earlier death and risk of stroke so additional evidence and discussion of power are needed.

The authors conduct several sex-specific tests across multiple outcomes and should adjust p-value significance thresholds for multiple hypothesis testing.

There are huge gradients in smoking by education and birth cohort. These variables are also highly correlated with life expectancy. The authors either need to control for education and birth cohort in their models or provide a strong justification for why these controls are not included.

The authors do not address how mortality selection affects their results. For example, it’s possible that only the healthiest male smokers survived past age 50 and this explains why we see an earlier onset of death for female smokers. This is a major limitation of the HRS data that at a minimum needs to be discussed if not corrected for somehow using a weighting strategy (e.g., see Domingue et al., 2017). Because of this, I think it would also strengthen the paper immensely if the authors were able to replicate their findings in another dataset that is slightly younger (e.g., UK Biobank).

[Domingue, B. W., D. W. Belsky, A. Harrati, D. Conley, D. R. Weir, and J. D. Boardman. 2017. "Mortality selection in a genetic sample and implications for association studies." International Journal of Epidemiology 46 (4):1285-1294.]

Along these lines, I would like to know which HRS cohorts were used in the study. For example, respondents in the AHEAD cohort were over the age of 70 when they entered the HRS. Including these cohorts in the analysis may exacerbate any bias from mortality selection.

It was unclear to me why active female smokers were more likely to have a stroke whereas passive male smokers were more likely to have a stroke, and I did not understand the authors’ rational behind these findings on p. 10 (“Thus it is not surprising the main-stream and second-hand smoke would diverge in toxicity, and have different gender interactions”). Why would we expect level of toxicity to affect sex differences in an outcome? Clarification and more discussion are needed here.

I like how the authors cite findings from animal models to provide some biological basis for their findings. However, the findings from mouse models that are reported do not seem to match the findings of the study. The authors claim that this contradiction is a result of the mouse models not representing “older human ages.” Please elaborate or provide more evidence for why the biological process of aging may change the sex-specific interactions we observe.

I like how the authors present findings for active and passive smoking exposure, and I thought the way they used the HRS data to calculate pack years was an excellent utilization of the data that’s available. However, if I remember correctly, there is a lot of missingness for the age of onset and cessation questions in the HRS. How many respondents in the study have imputed data for pack years? How might this affect the results?

In Table S1, the authors report that 80.3% of males and 73.2% of females are active smokers in the HRS. These percentages seem way too high and I would double-check these numbers (other publications report closer to 25% for males and 24% for females in the HRS).

In the HRS, former smokers are asked their max CPD over their entire smoking history and current smokers are asked their current CPD. Since individuals tend to ramp down their average daily number of cigarettes as they age, I would guess that active smokers on average have a lower CPD than former smokers. The authors should discuss how this reporting difference in the HRS may affect their overall results.

Minor Concerns:

On page 3, I found the sentence “this study highlights the biological basis of gender differences in smoking effects” to be misleading. This study does not explore the actual biological underpinnings of sex differences in smoking comorbidities.

Throughout, the authors use the word gender. If the hypothesis is that observed population-level differences have a biological basis I would use the word sex instead of gender.

I may have missed it, but what methods were used for imputation of the pack years variable?

Could the passive smoking variable also be calculated in pack years? If you know how long an individual has been married to their spouse it seems like you could multiply their spouse’s average packs per year times the number of years they’ve been married. Not sure if this would work in the HRS, but it seems like these findings may also be dose dependent.

The authors should provide a discussion of the limitations of their study.

The figures were fuzzy and I had difficulty reading them. Please provide higher quality files in future revisions.

6. PLOS authors have the option to publish the peer review history of their article (what does this mean?). If published, this will include your full peer review and any attached files.

Reviewer #1: No

Reviewer #2: No

---

## [Author Response · Author response to Decision Letter 0]

7 Apr 2020

Reviewer #1: 

Thank you for giving me the chance to review this manuscript.

The study investigated the association between smoking and different health outcomes in a large population-based sample. Furthermore, sex-specific and dose-dependent smoking effects were evaluated. The paper is well written and contributes with interesting results to the previous literature. However, some points need to be clarified or could be discussed in more detail. Please see below for my specific comments.

Specific comments

1. It could be mentioned in the abstract which statistical methods have been applied. 

Answer: Information on statistical methods was added to the abstract. 

2. In the abstract, HR 0.44-0.56 for the risk of lung disorders in women smokers is stated. This needs to be checked. 

Answer: We apologize for the confusion. The sentence was revised for clarification, and in order to refer to the main finding of sex differences found. “Women had earlier age of onset for lung disorders (HR=2.83, 95%CI:1.74-4.6), but men risk due to smoking were higher (Smoking-Sex interaction P<0.02) than women". 

3. What was the reason to exclude asthma? Only lung disorders such as chronic bronchitis and emphysema were considered.

Answer: Data for asthma was not available in HRS. HRS asked about a limited number of diseases or conditions regarded as having age-related onset. 

4. It is mentioned in the methods section that 12 waves of the Health and Retirement Study (1992-2014) were used for the current analyses. Does this imply that 12 follow-ups were available? Maybe some more information on the different waves and times of follow-up could be given.

Answer: New figures that summarize the number of individuals in each cohort, year of the entry, and number of years of follow-up were added to the supplementary data (Fig. S1-3). The HRS began in 1992 and 1993 with a cohort born from 1931 to 1941 (original HRS); subsequently 5 different cohorts have been added ending with the mid baby boomers. The HRS cohort can have up to 12 follow-up interviews and those who were the youngest, added in 2010, could only have 2 follow-ups. Since, most of the AHEAD cohort was entered at ages older than 70, they were predominantly excluded from the analysis. But a small number of AHEAD respondents who were age eligible were included. We have clarified this in the text.

5. Would it be possible to specify the cause of death? Besides smoking there might be other risk factors which could lead to an early death. Have further potential confounding factors such as socio-economic status, comorbidities and other lifestyle-related factors been tested in the regression models?

Answer: Unfortunately, we do not have access to cause of death data to answer the reviewer question. However, analysis of multiple health outcomes allows us to conclude that cigarette smoke can increase the risk of death from multiple chronic diseases. 

Because it is possible that other variables confound the analysis, we have added additional analysis including years of education and cohort as potential confounders of the association between smoking and health outcomes (Table S2-3). Controlling for these variables did not alter our original results or conclusions. We have added this to the 2nd to last paragraph in the discussion. 

6. It is stated in the methods section that missing smoking values were imputed using average values. What was the proportion of imputed values? If the proportion was quite high it would be good to perform sensitivity analyses only including those with self-reported smoking information (i.e. exclusion of subjects with imputed values).

Answer: We appreciate the reviewer’s comment and add information about data imputation in the supplementary data. The sensitivity analysis (Table S2) shows the same smoking-sex interaction as discussed in the paper. The analysis based on non-imputed data showed stronger p-values for sex-smoking interaction but did not change any of the conclusions or main findings. Thus, we report and refer to those results in the supplement. 

7. The exact definition of the smoking categories is not entirely clear (presented in Table S1). The percentages for never, passive and active smokers sum up to 100% for men and women, respectively, meaning that these three groups are disjunctive. However, passive smokers are per definition also never smokers. Does this imply that the never smoker group comprises only never smokers without passive smoke exposure? It seems that former smokers are included in the active smoking category. Have it been investigated whether there are different health effects when differentiating former (e.g. quit smoking a long time ago) and current smokers?

Answer: The demographic table was updated for clarification. “Active smokers” was replaced by “Ever smokers” in Table S1. 

We took advantage of the opportunity and added additional analysis about effects of smoking cessation effects as suggested by the reviewer to the results section, Table 3, figure 3, and additional discussion points. Only former smokers who quit smoking for more than 15 years partially had decrease of hazards rate for health outcomes than active smokers. 

8. The resolution of the Kaplan-Meier curve is not sufficient as the legend is not readable.

Answer: We apologize for this. It appears that the PlosOne website reduced the quality of the figures during conversion to PDF. We replaced the figures as suggested. Hopefully, they will have better quality in the revision. Please download the image separately from the link on the corner of the PDF file to see the image in original resolution. 

9. Potential biological mechanisms behind the sex-specific smoking effects could be discussed in more detail.

Answer: We have added to our discussion (2nd paragraph) the potential role of sex hormones in environmental responses. 

 

Reviewer #2: 

This study examines sex-specific vulnerability to health outcomes from smoking at older ages in the HRS. They find that older women are more susceptible to earlier death and risk of stroke but are less likely to report a doctor-diagnosed lung disorder. Sex-specific incidence and comorbidity of smoking and other substance use disorders is critically under-examined in population studies, and the paper makes a novel and important contribution to this literature. However, I have some major and minor concerns that need to be addressed before I can recommend it for publication. These concerns are highlighted below.

Major Concerns:

1. My main concern is whether the study is well powered, particularly for the sex-specific findings for females in the “very high pack years” group who comprise only 13.3% of female smokers. These findings form the basis of the authors’ claim that older female smokers are more susceptible to earlier death and risk of stroke so additional evidence and discussion of power are needed.

Answer: We understand the reviewer’s concern. As shown in Table S1 of the supplement, there was sufficient sample size throughout the range of levels of smoking. The 13.3% of females who are classified as very high smokers consist of 1560 women. This is a large number of individuals with up to 22 years of follow up after entry into the study of this population-representative sample. We added additional discussion points on the interaction of age and cigarettes, and the role of sex hormones in response to cigarette smoke for better clarification and interpretation on sex-specific findings. 

2. The authors conduct several sex-specific tests across multiple outcomes and should adjust p-value significance thresholds for multiple hypothesis testing.

Answer: Since, these are independent outcomes examining a pattern of morbidity and mortality with aging, we did not adjust p-values for multiple test corrections. However, most of the findings are quite strong and persist with multiple test correction for 5 interaction tests. Thus, we emphasize the pattern of effects and report the p-values, so that readers can extract the most important differences and extrapolate the findings to those relevant to other research. 

3. There are huge gradients in smoking by education and birth cohort. These variables are also highly correlated with life expectancy. The authors either need to control for education and birth cohort in their models or provide a strong justification for why these controls are not included.

Answer: Additional analysis was added to Table S3-4 to examine whether the inclusion of years of education and birth cohort would change the association between smoking, sex and health outcomes. The analysis revealed that these confounders do not alter the smoking hazards or sex-interactions. As mentioned in our response to reviewer 1, comment 5, we added additional discussion points to the limitations about disentangling the effects of confounders on smoking hazards. 

4. The authors do not address how mortality selection affects their results. For example, it’s possible that only the healthiest male smokers survived past age 50 and this explains why we see an earlier onset of death for female smokers. This is a major limitation of the HRS data that at a minimum needs to be discussed if not corrected for somehow using a weighting strategy (e.g., see Domingue et al., 2017). Because of this, I think it would also strengthen the paper immensely if the authors were able to replicate their findings in another dataset that is slightly younger (e.g., UK Biobank).

Answer: We thank the reviewer for this suggestion. A discussion paragraph was added to the paper about the potential selection effects resulting from lack of survival in HRS. Our sample includes primarily people who joined the HRS in their 50s, rather than those who joined at older ages (see supplemental Figures S1-S3). Our sample is different from the HRS genetic data that was used in the Domingue et al., 2017 paper. One of the major issues in that paper was that people who joined in 1992 had to survive to 2006 or 2008 to be included in the genetic data. 

Mortality before age 50 from smoking is fairly unusual and studying mortality among people in this age would be a very selected group. Studying smoking effects in younger ages is outside the scope of our paper. 

5. Along these lines, I would like to know which HRS cohorts were used in the study. For example, respondents in the AHEAD cohort were over the age of 70 when they entered the HRS. Including these cohorts in the analysis may exacerbate any bias from mortality selection.

Answer: The reviewer is correct in indicating we should have made this clear in the earlier draft. We have added material in the text and supplementary figures to clarify the sample used (Figures S1-S3). We did not include the AHEAD in this analysis. 

6. It was unclear to me why active female smokers were more likely to have a stroke whereas passive male smokers were more likely to have a stroke, and I did not understand the authors’ rational behind these findings on p. 10 (“Thus it is not surprising the main-stream and second-hand smoke would diverge in toxicity, and have different gender interactions”). Why would we expect level of toxicity to affect sex differences in an outcome? Clarification and more discussion are needed here.

Answer: We expanded this discussion on the chemical differences between main- and side-stream smoke (page 12) to answer this question. 

7. I like how the authors cite findings from animal models to provide some biological basis for their findings. However, the findings from mouse models that are reported do not seem to match the findings of the study. The authors claim that this contradiction is a result of the mouse models not representing “older human ages.” Please elaborate or provide more evidence for why the biological process of aging may change the sex-specific interactions we observe.

Answer: We added additional discussion points and now describe how age can alter the sex differences in cigarette toxicity (beginning on page 13). We highlighted how sexual hormones can affect the biological responses to the environmental stressors. Thus, sex-cigarette smoke interaction can differ at different life stages (e.g. pre and post menopause). 

8. I like how the authors present findings for active and passive smoking exposure, and I thought the way they used the HRS data to calculate pack years was an excellent utilization of the data that’s available. However, if I remember correctly, there is a lot of missingness for the age of onset and cessation questions in the HRS. How many respondents in the study have imputed data for pack years? How might this affect the results?

Answer: We added supplementary information on the imputation approach used and provide information on how much data were imputed. Comparative analysis using all the data and the data with no imputation was added to the supplement, Figure S5. The analysis based on non-imputed data actually showed stronger p-values for sex-smoking interaction, but did not change any of the conclusions or main findings. Thus, we report the unimputed results in the supplement and explain this in the main paper. 

9. In Table S1, the authors report that 80.3% of males and 73.2% of females are active smokers in the HRS. These percentages seem way too high and I would double-check these numbers (other publications report closer to 25% for males and 24% for females in the HRS).

Answer: We apologize for using the word “active”. This was a mistake as this number includes both current and former smokers so these would be “ever” or current smokers in conventional terminology. The table was revised for clarification. 

10. In the HRS, former smokers are asked their max CPD over their entire smoking history and current smokers are asked their current CPD. Since individuals tend to ramp down their average daily number of cigarettes as they age, I would guess that active smokers on average have a lower CPD than former smokers. The authors should discuss how this reporting difference in the HRS may affect their overall results.

Answer: As the reviewer suggested, we added a Figure S4 to compare reports of the number of cigarettes per day between current and former smokers. It is true that there is a modest excess CPD in former smokers. We include this as a limitation in the discussion. 

Minor Concerns:

11. On page 3, I found the sentence “this study highlights the biological basis of gender differences in smoking effects” to be misleading. This study does not explore the actual biological underpinnings of sex differences in smoking comorbidities.

Answer: The sentence was modified. 

12. Throughout, the authors use the word gender. If the hypothesis is that observed population-level differences have a biological basis I would use the word sex instead of gender.

Answer: Gender was replaced by sex throughout the text. 

13. I may have missed it, but what methods were used for imputation of the pack years variable?

Answer: We apologize if the text was not clear. We have expanded the methods section to describe the imputation method. In addition, Figure S5 and Table S2 were added to the supplement providing details on the imputation and sensitivity analysis using only the non-imputed sample. 

14. Could the passive smoking variable also be calculated in pack years? If you know how long an individual has been married to their spouse it seems like you could multiply their spouse’s average packs per year times the number of years they’ve been married. Not sure if this would work in the HRS, but it seems like these findings may also be dose dependent.

Answer: This was our original intent. However, it was not possible to calculate the dosage of secondhand smoke reliably because of the large number of people with multiple spouses and varying lengths of marriage. 

15. The authors should provide a discussion of the limitations of their study.

Answer: We added further discussion of our limitations. 

16. The figures were fuzzy and I had difficulty reading them. Please provide higher quality files in future revisions.

Answer: We apologize for this. It appears that the PlosOne website reduced the quality of the figures during conversion to PDF. We replaced the figures as suggested. Hopefully, they will have better quality in the revision. Please download the image separately from the link on the corner of the PDF file to see the image in original resolution.

---

## [Decision Letter · Decision Letter 1]

14 May 2020

PONE-D-20-03538R1

Female vulnerability to the effects of smoking on health outcomes in older people

PLOS ONE

Dear Dr. Haghani,

Thank you for submitting your manuscript to PLOS ONE. After careful consideration, we feel that it has merit but does not fully meet PLOS ONE’s publication criteria as it currently stands. Therefore, we invite you to submit a revised version of the manuscript that addresses the points raised during the review process.

More specifically, we ask that you carefully consider Reviewer 1's comments about definitions and analytic suggestions. 

We would appreciate receiving your revised manuscript by Jun 28 2020 11:59PM. To enhance the reproducibility of your results, we recommend that if applicable you deposit your laboratory protocols in protocols.io, where a protocol can be assigned its own identifier (DOI) such that it can be cited independently in the future. For instructions see: http://journals.plos.org/plosone/s/submission-guidelines#loc-laboratory-protocols

We look forward to receiving your revised manuscript.

Kind regards,

Neal Doran

Academic Editor

PLOS ONE

Reviewers' comments:

Reviewer's Responses to Questions

**Comments to the Author**

1. If the authors have adequately addressed your comments raised in a previous round of review and you feel that this manuscript is now acceptable for publication, you may indicate that here to bypass the “Comments to the Author” section, enter your conflict of interest statement in the “Confidential to Editor” section, and submit your "Accept" recommendation.

Reviewer #1: All comments have been addressed

Reviewer #2: All comments have been addressed

2. Is the manuscript technically sound, and do the data support the conclusions?

Reviewer #1: Partly

Reviewer #2: Yes

3. Has the statistical analysis been performed appropriately and rigorously? 

Reviewer #1: Yes

Reviewer #2: Yes

4. Have the authors made all data underlying the findings in their manuscript fully available?

Reviewer #1: Yes

Reviewer #2: Yes

5. Is the manuscript presented in an intelligible fashion and written in standard English?

Reviewer #1: Yes

Reviewer #2: Yes

6. Review Comments to the Author

Reviewer #1: I thank the authors for the thorough revision of the manuscript and conducting several additional analyses. I just have one further comment regarding the analyses on smoking cessation. Investigating sex-specific effects of smoking cessation on different health outcomes might be beyond the scope of this manuscript and its aims. As in the first version of the manuscript the exact definition of “active smokers” (now indicated as “ever smokers”) was not entirely clear and the assessment of lifetime pack-years smoked differs by smoking status (average number of cigarettes per day vs. maximum number of cigarettes), sensitivity analyses investigating whether the effects of smoking on health outcomes differ by smoking status might have been reasonable. Therefore, ever smokers could have been separated into former and never smokers (taking lifetime pack-years into account) independent of years since smoking cessation. Table 3 in the manuscript shows some quite high hazard ratios with broad confidence intervals. This might be due to the small sample sizes, esp. for women who quit smoking <5 or 5-15 years ago and who have lung disorders. Thus, these results should be interpreted with caution. I would suggest to move these results to the supplement and not put too much emphasis on this kind of analyses. Rather, I would just add this as kind of sensitivity analyses (as was done e.g. for the comparison of imputed vs. non-imputed pack-years data) to see whether the effects change and just mention this briefly. Nevertheless, I would like to thank the authors for all their efforts.

Reviewer #2: (No Response)

7. PLOS authors have the option to publish the peer review history of their article (what does this mean?). If published, this will include your full peer review and any attached files.

Reviewer #1: No

Reviewer #2: No

---

## [Author Response · Author response to Decision Letter 1]

14 May 2020

Reviewer #1: I thank the authors for the thorough revision of the manuscript and conducting several additional analyses. I just have one further comment regarding the analyses on smoking cessation. Investigating sex-specific effects of smoking cessation on different health outcomes might be beyond the scope of this manuscript and its aims. As in the first version of the manuscript the exact definition of “active smokers” (now indicated as “ever smokers”) was not entirely clear and the assessment of lifetime pack-years smoked differs by smoking status (average number of cigarettes per day vs. maximum number of cigarettes), sensitivity analyses investigating whether the effects of smoking on health outcomes differ by smoking status might have been reasonable. Therefore, ever smokers could have been separated into former and never smokers (taking lifetime pack-years into account) independent of years since smoking cessation. Table 3 in the manuscript shows some quite high hazard ratios with broad confidence intervals. This might be due to the small sample sizes, esp. for women who quit smoking <5 or 5-15 years ago and who have lung disorders. Thus, these results should be interpreted with caution. I would suggest to move these results to the supplement and not put too much emphasis on this kind of analyses. Rather, I would just add this as kind of sensitivity analyses (as was done e.g. for the comparison of imputed vs. non-imputed pack-years data) to see whether the effects change and just mention this briefly. Nevertheless, I would like to thank the authors for all their efforts.

Answer: Thank you for this comment. As suggested, table 3 was moved to the supplement. We also softened the language about the conclusions on sex differences in the benefits of smoking cessation.

---

## [Editor Report · Decision Letter 2]

18 May 2020

Female vulnerability to the effects of smoking on health outcomes in older people

PONE-D-20-03538R2

Dear Dr. Haghani,

We are pleased to inform you that your manuscript has been judged scientifically suitable for publication and will be formally accepted for publication once it complies with all outstanding technical requirements.

With kind regards,

Neal Doran

Academic Editor

PLOS ONE
---

## [Editor Report · Acceptance letter]

22 May 2020

PONE-D-20-03538R2 

Female vulnerability to the effects of smoking on health outcomes in older people 

Dear Dr. Haghani:

I am pleased to inform you that your manuscript has been deemed suitable for publication in PLOS ONE. Congratulations! Your manuscript is now with our production department. 

With kind regards,

on behalf of

Dr. Neal Doran 

Academic Editor

PLOS ONE